# Safety and Efficacy upon Infection in Sheep with Rift Valley Fever Virus ZH548-rA2, a Triple Mutant Rescued Virus

**DOI:** 10.3390/v16010087

**Published:** 2024-01-05

**Authors:** Sandra Moreno, Gema Lorenzo, Álvaro López-Valiñas, Nuria de la Losa, Celia Alonso, Elena Charro, José I. Núñez, Pedro J. Sánchez-Cordón, Belén Borrego, Alejandro Brun

**Affiliations:** 1Centro de Investigación en Sanidad Animal (CISA), Instituto Nacional de Investigación y Tecnología Agraria y Alimentaria, Consejo Superior de Investigaciones Científicas (INIA-CSIC), Valdeolmos, 28130 Madrid, Spain; sandra.moreno@csic.es (S.M.); lorenzo.gema@csic.es (G.L.); pedrojose.sanchez@inia.csic.es (P.J.S.-C.); 2Centre de Recerca en Sanitat Animal (CReSA), Institut de Recerca i Tecnologia Agroalimentàries (IRTA), Bellaterra, 08193 Barcelona, Spainjoseignacio.nunez@irta.cat (J.I.N.)

**Keywords:** Rift Valley fever virus, attenuating mutations, vaccine efficacy, sheep pathology

## Abstract

The introduction of three single nucleotide mutations into the genome of the virulent RVFV ZH548 strain allows for the rescue of a fully attenuated virus in mice (ZH548-rA2). These mutations are located in the viral genes encoding the RdRp and the non-structural protein NSs. This paper shows the results obtained after the subcutaneous inoculation of ZH548-rA2 in adult sheep and the subsequent challenge with the parental virus (ZH548-rC1). Inoculation with the ZH548-rA2 virus caused no detectable clinical or pathological effect in sheep, whereas inoculation of the parental rC1 virus caused lesions compatible with viral infection characterised by the presence of scattered hepatic necrosis. Viral infection was confirmed via immunohistochemistry, with hepatocytes within the necrotic foci appearing as the main cells immunolabelled against viral antigen. Furthermore, the inoculation of sheep with the rA2 virus prevented the liver damage expected after rC1 virus inoculation, suggesting a protective efficacy in sheep which correlated with the induction of both humoral and cell-mediated immune responses.

## 1. Introduction

Rift Valley fever (RVF) is a viral zoonotic disease that affects domestic and wild ungulates. The disease leads to high abortion rates in herds and neonatal mortality, making it a significant threat to livestock and public health. The disease has a significant economic impact, particularly in rural communities that depend on local livestock for their livelihoods, especially in developing countries. The causative virus, RVFV, is transmitted via the bite of mosquitoes, mainly of the genera *Aedes* and *Culex*. The disease is currently endemic in many countries of sub-Saharan Africa, the Middle East and Indian Ocean islands, and numerous cases of seropositive animals have been reported in North African countries. Humans can be severely affected by the disease if exposed to infected mosquitoes, contaminated food or animal byproducts. Though the disease has been fundamentally restricted to the African continent, the spread of the virus is also possible via several means, including the import of viremic livestock [1] or via humans returning to their home countries from RVF endemic areas [2].

RVFV belongs to the order *Bunyavirales*, the family *Phenuiviridae* and the genus Phlebovirus. At least 15 different genetic lineages have been described [3], but for the moment, only one serotype is recognised. Structurally, the RVFV virions are icosahedral, with a diameter of approximately 80–100 nm [4], composed of heterodimers of two membrane glycoproteins (Gn and Gc) [5,6,7], enclosing a nucleocapsid that packs at least three single-stranded RNA segments of different sizes (L, M, S), and negative and ambisense polarity [8]. The large (L) RNA segment encodes the RNA-dependent RNA polymerase (RdRp) [9,10], while the medium (M) segment encodes a polyprotein that, upon protease processing, gives rise to the two membrane glycoproteins (Gn and Gc) [11] and a non-structural protein (NSm) related to apoptosis modulation [12]. The messenger RNA of the M segment can translate at least an additional protein of 78 kDa protein comprising the NSm and Gn ORFs [13]. Finally, the small (S) segment encodes the virus nucleoprotein N and the virulence-associated factor NSs in opposite orientations, separated by an intergenic region (IGR) [14,15,16].

An effective method to prevent the potential spread of RVF is vaccination, and to date, three types of vaccines have been licensed for use in livestock, both attenuated and inactivated [17]. There are also several experimental vaccines based on the rational deletion of non-structural genes associated with virulence or on mutagenisation [18]. In many cases, these vaccines retain some residual virulence when tested in more sensitive animal models or in early-stage pregnant animals, so there is still room for the improvement of their safety profile.

As a result of previous research, a mutagenised variant of the South African 56/74 virus isolate was obtained via cell culture propagation in the presence of favipiravir, a purine nucleoside analogue with antiviral properties whose incorporation into nascent RNA strands causes base mispairing and increases the mutation rate of the virus. After the eighth serial passage in the presence of a suboptimal dose of favipiravir (40 µM), a resistant virus variant was selected. More specifically, sequence analysis showed the inclusion of a total of 47 base mutations along the three RNA segments, 23 of which were non-synonymous (NS) changes. Most NS mutations (14) accumulated in the M-segment encoding the GnGc polyprotein, 7 in the viral polymerase gene and 2 in the non-structural protein NSs [19]. The most interesting finding was the highly attenuated nature of the resulting virus, even in immunodeficient A129 (IFNARKO) mice. Accordingly, we proposed the name “hyper attenuated RVFV variant 40Fp8” [20].

The structure of the RVFV viral polymerase has recently been described [21] and contains all the catalytic motifs described for negative-polarity RNA virus RNA polymerases [22]. To characterise the role of key mutations in the attenuation of the RVFV-40Fp8 virus, we focused first on those mutations that could be related to favipiravir resistance, which intuitively should correspond to the viral polymerase (RdRp). Three mutations found in the catalytic core of the 40Fp8 RdRp were close to these canonical motifs responsible for the entry of template RNA, nucleotides, and metal ions, which cooperate together for polymerisation. Residues 924 and 1303 are highly conserved among phleboviruses, and the introduction of G924S and A1303T mutations into the viral RdRp of the ZH548 RVFV strain was enough to fully attenuate the virus, as demonstrated upon intraperitoneal inoculation into mice, indicating its predominant role in virus attenuation [23].

As for the two mutations found in the S segment affecting the non-structural protein NSs, known as the main virulence factor, we focused on the P82L mutation because it constitutes a drastic change and is also found in one of the proline-rich areas previously linked to the ability of NSs to form typical intranuclear filaments in infected cells [24]. In this case, the introduction of P82L into the NSs protein of ZH548 resulted in partial attenuation of the virus when inoculated into mice [25]. As expected, a triple mutant G924S, A1303T, P82L virus (termed ZH548-rA2) was also highly attenuated. The results obtained in mice prompted us to test whether this level of attenuation would be maintained in the natural target species, as well as the ability of the triple mutant to induce protection upon challenges from the parental virus. Here, we show that these positions can be targeted to increase the safety of attenuated RVF vaccines in livestock.

## 2. Materials and Methods

### 2.1. Cells and Viruses

The Vero cell line (African green monkey kidney cells, ATCC CCL-81) was used for this study. Cells were grown in Dulbecco’s modified Eagle medium (DMEM, Bio-West, Nuaillé, France) supplemented with 10% foetal bovine serum (Gibco™-FBS, Thermo Fisher Sci. Waltham, MA, USA), 2 mM L glutamine, 1% 100× non-essential amino-acids and 100 U/mL penicillin/100 μg/mL streptomycin. All cells were incubated at 37 °C in the presence of 5% CO_2_. The viruses used in this work are the rescued ZH548 (wild-type, short named as rC1) and the ZH548 triple mutant virus (short named as rA2) carrying G924S and A1303T mutations in the L-polymerase and a full-length P82L mutated NSs protein as described [23]. Infections and titrations were performed as described [20].

### 2.2. Animal Study Design

A total of six, 4-month-old, female Churra sheep (Spanish native breed), weighing about 30–40 kg, were supplied by a farm with a high sanitary status. The sheep were housed at the CISA-BSL3 facilities and acclimatised for one week before the commencement of the experiment. In addition to a straw bed, animals were provided with water and food *ad libitum.* The animals were randomly divided into three groups, each consisting of two animals. The day of experimental inoculation was designated as day 0. Before inoculation, all animals were bled. Throughout the experiment, rectal temperatures and clinical signs were monitored daily. In group 1 (G1), animals were inoculated subcutaneously in the neck area near the pre-scapular lymph node with 1 mL containing a high dose of ZH548-rA2 virus (10^7^ pfu). Whole blood samples were taken from the jugular vein at 0, 3, 4, 5, 6 and 7 days after inoculation (dpi). At 7 dpi, both animals were euthanised and necropsied to assess macroscopic lesions. Furthermore, the immunostimulatory capacity (i.e., the induction of humoral or cellular responses) and protective efficacy of the rA2 virus was assessed in another two sheep (G2). Three weeks after the inoculation with ZH548-rA2, following the same procedure described for G1, animals were challenged by the subcutaneous route with 1 mL containing 10^7^ pfu of the rescued parental virus (ZH548-rC1). Sheep belonging to G2 were sampled at 0, 1, 2, 3, 4, 5, 7, 14, and 21 dpi. A control group of two non-immunised sheep (G3) was also infected subcutaneously with 1 mL containing 10^7^ pfu of the rC1 virus. After challenge, blood samples were taken at 0, 1, 2, 3 and 4 dpi from G2 and G3 sheep until they were euthanised and necropsied between days 3 and 4 post challenge (dpc). Animal experiments were performed following the EC guidelines (Directive 86/609) upon approval by the Ethical Review and Animal Care Committees of INIA and Comunidad de Madrid (authorisation decision PROEX 192/17).

### 2.3. Blood Chemistry

Whole blood samples from sheep in G2 and G3 were collected in BD Vacutainer tubes and centrifuged at 1267× *g* for 10 min for serum and/or plasma separation. Serum and plasma samples were stored at −80 °C until use. The samples were then tested with a VetScan VS2 analyser using a large animal profile reagent rotor (Abaxis, Union City, CA, USA) for the measuring of several parameters including ALB, ALP, AST, BUN, Ca, CK, GGT, GLOB, Mg, PHOS, and TP levels.

### 2.4. Virus Detection in Blood Samples

RNA was extracted from EDTA blood samples collected at the indicated time points using an RNA virus extraction kit (Speedtools, 180 Biotools BM, Madrid, Spain) as described [26]. Viremia levels were then assessed using a real-time RT-qPCR specific to the RVFV L-segment [27]. The correspondence between the Cq values and the plaque forming units was estimated using sheep blood spiked with 10^1^ to 10^5^ pfu of a RVFV virus stock previously titrated via plaque assay. For virus isolation, 100 μL of defibrinated blood or plasma were diluted with DMEM and added onto freshly plated Vero cell monolayers for one hour in a 5% CO_2_ incubator at 37 °C. Upon extensive washing, the cultures where further incubated for 5–6 days. Cell lysates were generated after three consecutive freeze-thawing cycles, and clarified supernatants were incubated onto fresh Vero cell monolayers for a total of three rounds.

### 2.5. Sequencing Analysis

The whole genome sequence of the rA2 virus was determined by next-generation sequencing (NGS). Briefly, RNA extracted from the supernatant of infected cells was used as a template for Double Strand cDNA synthesis using the PrimeScript™ Double Strand cDNA Synthesis Kit (Takara Bio, Kusatsu, Japan) following the manufacturer’s instructions. A pool of 10 RVFV specific primers covering the whole viral genome was used [20]. After analysis and quantification of the final product using a high sensitivity D5000 Screen Tape (Agilent, Santa Clara, CA, USA), libraries were generated using the NEBNext^®^ Ultra™ II DNA Library Prep Kit (New England Biolabs, Ipswich, MA, USA) and sequenced with an MiSeq instrument, using protocols and reagents from Illumina (Illumina^®^, San Diego, CA, USA) at the Unidad Genómica FPCM (Madrid, Spain). An analysis of the reads was performed as described [23,28]. The readings were aligned against the available sequences corresponding to the 3 segments of RVFV strain ZH548 (accession numbers DQ375403 (L), DQ380206 (M) and DQ380151(S) [29,30]).

### 2.6. Tissue Sampling, Histopathological and Immunohistochemical Analysis

During necropsies, macroscopic evaluations of the lesions were performed. In addition, different tissue samples were fixed via immersion in 4% buffered formalin solution for 72 h, routinely processed and embedded in paraffin wax. Furthermore, 4 µm tissue sections were stained with haematoxylin and eosin (H&E) and then microscopically evaluated.

To visualise the viral antigen, immunohistochemistry was performed on tissue sections. Briefly, the sections were deparaffinised and rehydrated. Endogenous peroxidase activity was inhibited via incubation with 3% hydrogen peroxide in methanol for 30 min at room temperature (RT). The samples were then subjected to heat-induced epitope retrieval via immersion in citrate buffer (pH 3.2) at sub boiling for 10 min. After treatment, the sections were cooled to RT and rinsed in TBS buffer (pH 7.2) for 5 min. Nonspecific binding sites were blocked with 5% normal goat serum in TBS buffer for 30 min. After the blocking step, the sections were incubated with an in-house polyclonal rabbit anti-RVFV serum (diluted 1:500–1:1000 in TBS buffer) overnight at 4 °C. Next, the sections were washed with TBS 1×—Tween20 (0.2%) (1 × 5 min) and TBS (2 × 5 min), followed by 1 h incubation at RT with a secondary antibody (EnVision labelled polymer goat anti-rabbit, DAKO, Glostrup, Denmark). A working solution of 3,3′-diaminobenzidine tetrahydrochloride (DAB) (DAKO REAL EnVision Detection System) was applied as the chromogen for 45 s to the slides to be stained. The slides were rinsed with distilled water, counterstained with Mayer’s haematoxylin, dehydrated and mounted. Tissue samples from infected and non-infected sheep were included as test controls. In addition, an in-house irrelevant rabbit antiserum was used as the primary antibody negative technique control.

### 2.7. Expression of Recombinant Antigens

The gene corresponding to the RVFV ZH548 Gn glycoprotein ectodomain sequence (encoding amino acids 154–580) was cloned into a pFastBac HT donor transfer vector and used to transform DH10B Bac *E. coli* cells (Bac-to-Bac expression system, Thermo Fisher, Waltham, MA, USA). The recombinant baculovirus generated was used to infect HighFive cells (H5). After three days, the infected cells were collected, lysed and processed following established protocols. The Gn protein was purified under denaturing conditions using HisTrap FF columns (GE Healthcare, Chicago, IL, USA) to the Akta prime plus (GE Healthcare). Collected fractions were analysed via SDS-PAGE, pooled, dialysed against phosphate buffer and further concentrated (Vivaspin 6 100 kDa; GE Healthcare). Finally, the Gn protein was quantified using the BCA Protein Assay Kit (Pierce™, Waltham, MA, USA).

A RVFV N full-length construct was cloned via recombination (inPhusion cloning, Promega, Madison, WI, USA) into the pOPIN-E vector (Protein Production Facility, Oxford, UK), which adds a C-terminal hexahistidine tag. The expression of recombinant N in IPTG-induced *E. coli* BL21 cells was confirmed via SDS-PAGE and Western blot. A larger prep of affinity purified recombinant N protein was kindly provided by Dr Hani Boshra (Abrescia´s lab, CIC-BioGune, Derio, Spain) using a Ni-NTA column (GE Healthcare) followed by size-exclusion gel-filtration chromatography using a HiLoad Superdex 200 pg column (GE Healthcare).

### 2.8. Antibody Assays

A seroneutralisation assay was performed in multiwell 96 plates using sheep sera (in quadruplicates) that were 2-fold diluted starting from 1/10, mixed with an equal volume of infectious virus containing 100 TCID_50_ and incubated for 30 min at 37 °C. Then, a Vero cell suspension was added, and plates were incubated in a 5% CO_2_ atmosphere for 4 days. Monolayers were then monitored for the development of cytopathic effect (CPE), fixed with a 10% solution of formaldehyde, and stained with a 2% crystal violet solution. Neutralisation titres are expressed as the highest dilution of serum (in log_10_) rendering a 50% reduction in infectivity.

### 2.9. Interferon-γ Detection in Sheep Plasma and Upon In Vitro Re-Stimulation Analysis

An antibody sandwich ELISA using a capture anti-bovine IFNγ antibody was performed as described [31]. Furthermore, 100 µL of fresh plasma samples from sheep in groups G2 and G3 collected after rA2 immunisation and after rC1 virus challenge were added to ELISA plate wells previously coated with purified mAb anti-IFNγ (MT17.1, Mabtech, Nacka Strand, Sweden). Upon incubation and completing the washing steps, a detector, anti-bovine IFNγ mAb conjugated with biotin (MT307, Mabtech), was added. Streptavidin-HRPO (Becton–Dickinson, Eysins, Switzerland) was used for the detection of immunocomplexes upon the addition of the TMB peroxidase substrate (Sigma-Aldrich, San Luis, MO, USA). The absorbance values were determined at 450 nm using an automated reader (BMG, Labtech, Ortenberg, Germany). For the in vitro re-stimulation assay, 0.5 mL of whole blood samples were incubated in MW24 plate wells with 1 µg/mL of recombinant purified proteins Gn, or N, or 5µg of ZH548-Gn-derived peptides #19 (_203_-FQSYAHHRTLLEAVH-_217_) and #21 (_211_-TLLEAVHDTIIAKAD-_225_), or ZH548-Gc-derived peptides #226 (_1031_-AAFLNLTGCYSCNAG-_1045_) and 253 (_1139_-SWNFFDWFSGLMSWF-_1153_) added to each well and incubated at 37 °C in a 5% CO_2_ incubator for three days. The plasma was recovered from wells via plate centrifugation (300× *g*) and tested as described above in the IFNγ capture mAb sandwich assay.

## 3. Results

### 3.1. Clinical and Pathological Findings

To evaluate the safety and efficacy of the triple mutant virus ZH548-rA2 in sheep, one of the natural hosts of RVFV, a small pilot trial was conducted (Figure 1). After inoculation with the rA2 virus, none of the sheep included in the G1 and G2 groups showed any apparent signs of disease. Only a transient and mild increase in temperature at 1 dpi was recorded in sheep #73, #83, #39 and #40 (Figure 2A). A more evident increase in temperature was observed at 3 dpi in sheep included in G3 that received the same dose of the parental rC1 virus (#768, #909), with sheep #909 showing the highest temperature by 4 dpi. Despite the increase in temperature, no other clinical signs were evident. The challenge with the rC1 virus did not cause a temperature rise in sheep #39 and #40 (G2) that had received a previous inoculation with the rA2 virus (Figure 2B).

Since one of the main affected organs after a RVFV infection is the liver, different enzymes indicative of hepatic injury were evaluated after challenge with the rC1 virus via blood chemistry using serum samples (Figure 3). Aspartate amino transferase (AST) levels, normally present in low concentrations, were elevated in G3 sheep compared to G2 sheep, where levels remained normal. However, gamma-glutamyl transferase (GGT) levels remained elevated in both groups. No differences were found with respect to plasma protein degradation, as suggested by the detected levels of plasma albumin (ALB) or blood urea nitrogen (BUN), both indicative of liver or kidney disease. Additional biochemical parameters (see Materials and Methods) were measured and found to be within the normal ranges. 

During necropsies, none of the rA2-inoculated sheep included in either G1 or G2 showed any relevant macroscopic lesions. However, the livers of both animals inoculated with rC1 (G3) showed the most remarkable macroscopic findings characteristic of RVFV infection. The liver parenchyma showed a loss of consistency together with the presence of greyish-white foci, 0.5–2 mm in diameter, consisting of scattered necrotic lesions, many of them surrounded by a reddish halo. The number of foci was more abundant in sheep #909 necropsied on day 4 (Figure 4).

A histopathological evaluation of the livers of G3 sheep revealed findings characteristic of RVFV infection, with multifocal necrotic foci filled with cellular debris and both degenerated and viable neutrophils, surrounded by mononuclear cells, mainly lymphocytes. Hepatocytes with eosinophilic intranuclear inclusion bodies were also present. Viral infection was also confirmed via immunohistochemistry, with hepatocytes within the necrotic foci appearing as the main cells immunolabelled against viral antigen (Figure 5). Immunolabelled Kupffer cells and circulating monocytes were also occasionally observed in the hepatic sinusoids.

In contrast to what was observed in G3 sheep, histopathological evaluations revealed no liver lesions in any of the rA2-inoculated sheep (G1) euthanised on day 7 pi. Likewise, animals that were first immunised with rA2 and then challenged with the rC1 virus (G2) showed no liver injury (Figure 6). Collectively, all these data indicate that rA2 inoculation in sheep is safe and is able to avoid the pathology induced by the inoculation of rC1.

### 3.2. Analysis of Viremia Levels Via RT-qPCR and Virus Isolation

The presence of viral genomes was tested in blood samples taken at different times post inoculation. Consistent with the absence of pathology observed in the livers, no positive detection above the detection threshold was found in the blood of any of the rA2-inoculated sheep (G1 and G2). Attempts to isolate the virus upon blind passages in cell cultures also yielded negative results. Surprisingly, no viral genome was detected in the blood of rC1-infected sheep taken at 3 or 4 dpi despite the clear liver pathology observed both macro- and microscopically. Known amounts of the virus were spiked into sheep blood that rendered positive Cq, ruling out the possibility of PCR inhibition via some components upon the RNA extraction procedure. Attempts to isolate the virus from rC1 blood samples upon serial blind passage in cell culture were also negative. These data may indicate that the replication of ZH548-rescued viruses in sheep is limited, conditioning the molecular detection of viral genomes as well as viral isolation in cell cultures.

### 3.3. Analysis of Immune Responses

Upon rA2 virus inoculation (G2), neutralising antibodies were detected in serum samples from sheep #39 and #40 as early as day 4 post inoculation (pi), with a peak titre on day 21 pi. In the animals that were further challenged with the rC1 virus, a booster effect was observed in samples that were 3 days post challenge (Figure 7A). Neutralising antibodies were also tested and detected in sheep #73 and #83 (G1) on day 7 pi. Increased interferon gamma concentrations were also detected in plasma samples from sheep inoculated with the rC1 virus (G3) but not in those that had been previously inoculated (vaccinated) with the rA2 virus (G2) after rC1 virus challenge (Figure 7B). However, IFNγ secretion was re-stimulated in these sheep using recombinant Gn or N antigens in whole blood samples taken at 7, 14 and 21 days after rA2 inoculation (G2) (Figure 8A). The specificity of this result was further confirmed via the re-stimulation of sheep blood with four 12 mer peptides corresponding to Gn and Gc glycoproteins, indicating the presence of RVFV-specific IFNγ-secreting T cells (Figure 8B). These results support the proper induction of an immune response in sheep to levels that can be considered protective in spite of the clear attenuation of the rA2 virus.

## 4. Discussion

In a previous study [23] we showed that the triple mutant ZH548-rA2 virus was completely attenuated in mice. The high susceptibility of laboratory mice to virulent RVFV infection makes this infection model suitable for testing the efficacy of a vaccine approach. The inoculation of rA2-protected mice upon a lethal challenge with rC1 by means of a neutralising antibody response at varying levels correlated with protection. To confirm whether this positive result could be extrapolated to RVFV target species, we conducted a preliminary exploratory study in sheep, a model commonly used for RVF vaccine efficacy studies. To comply with animal welfare regulatory requirements, we minimised animal use by using only six animals, four of which received a dose of the rA2 virus for pathological and immunological assessments and four of which were challenged with the parental virus rC1 to test the protective effect of rA2 as a vaccine. Although the reduced sample size hampers statistical significance, the results obtained suggest that the rA2 virus in sheep retains the phenotype observed in mice, i.e., a marked attenuation compared to the parental virus and the ability to induce a protective immune response.

We used two rescued pure virus populations (rC1 and rA2) that were confirmed via NGS data, so that phenotypic differences could be attributed to only three amino acid residues. Our results showed that the rA2 virus was unable to cause pathology in sheep even at high doses of up to 10^7^ pfu. This suggests that the virus retains a high level of attenuation. An assessment of virus replication via RT-qPCR in blood samples suggests that rA2 is poorly disseminated since no virus isolation could be achieved after three blinded serial passages in cell culture. However, we cannot rule out a minimal level of replication as serum-neutralising antibodies were readily detected. Another interesting observation was the lack of detectable viremia in sheep inoculated with the rescued rC1 virus where, as with rA2, no virus could be isolated in cell culture. However, rC1 caused remarkable liver damage, also inducing elevated AST levels and increased plasma IFNγ levels, indicating an ongoing response to infection. One possibility is that rC1, derived from a pure rescued viral clone with reduced genetic diversity compared to a natural isolate, represents a virus with low pathogenicity and reduced ability to spread. It is important to note that in previous experiments in mice, rC1 was fully virulent [23] and liver damage was massive by day 3–4 post inoculation (unpublished observation). In sheep, the observation of scattered necrotic foci in the liver may be a consequence of better control of the viral spread once in the liver, preventing further systemic spread of the virus and precluding the detection of viremia. On the other hand, to our knowledge, the strain ZH548 has never been inoculated in sheep, so we have no previous data on the virulence of this human-origin isolate in sheep. Previous reports have suggested that the ZH548 virus isolate was less virulent than ZH501 [33], a contemporary isolate of identical lineage. Another possibility is that the virulent effect in sheep manifests after 4 days post inoculation, as the animals inoculated with rC1 were sacrificed on days 3 and 4 post challenge. It is conceivable that delayed viremia peaks may have impeded the early detection of viral genomes or isolation. As ZH548 has not been previously inoculated in sheep, its kinetics of infection were unknown. We posited that the progression of the infection in sheep with ZH548 would resemble that of other RVFV isolates, manifesting as peak viremia between two and three days post inoculation. Our findings indicate that this assumption needs to be revised. Nonetheless, the use of ZH548 for the challenge was solely conducted to enable a direct comparison with the rA2 virus and to attribute any differences in phenotype to the three specific mutations. It would be interesting to examine whether the rA2 virus offers protection against challenge with a heterologous isolate. Nonetheless, the exploratory nature of our preliminary experiment precluded the inclusion of more animals in the trial.

Nevertheless, inoculation with rA2 was able to induce neutralising antibodies, explaining the absence of liver lesions observed in vaccinated sheep challenged with the rC1 virus. Since the sole introduction of three amino acid changes caused this effect, it can be concluded that these three residues could be targeted for improvement of live attenuated RVFV vaccines, provided that they do not interfere with the growth or replication of the virus to the extent of compromising the induction of immune responses.

One of the best characterised RVF vaccine candidates is the mutagenised attenuated variant MP12. In MP12, the combination of three changes (Y259H (Gn), R1182G (Gc), and R1029K (L)) were sufficient to attenuate the ZH501 virus [34]. Therefore, a number of residue positions are being identified that could be targeted to improve the safety of live attenuated vaccine candidates generated via reverse genetics. It would be interesting to test, for instance, whether the attenuating mutations carried by rA2 would reduce the residual virulence of the MP12 or Clone13 vaccines. Both vaccines retain the ability to kill type-I Interferon receptor KO mice (A129) and can compromise the safety of ruminant foetuses in the initial steps of pregnancy when used at high doses [35,36]. Conversely, the introduction of MP12-attenuating mutations in the rA2 virus could perhaps reduce the virulence observed in A129 mice. Thus, the identification of attenuating mutations in the genome of RVFV constitutes a piece of valuable knowledge to increase the safety of RVF vaccines.

## Figures and Tables

**Figure 1 viruses-16-00087-f001:**
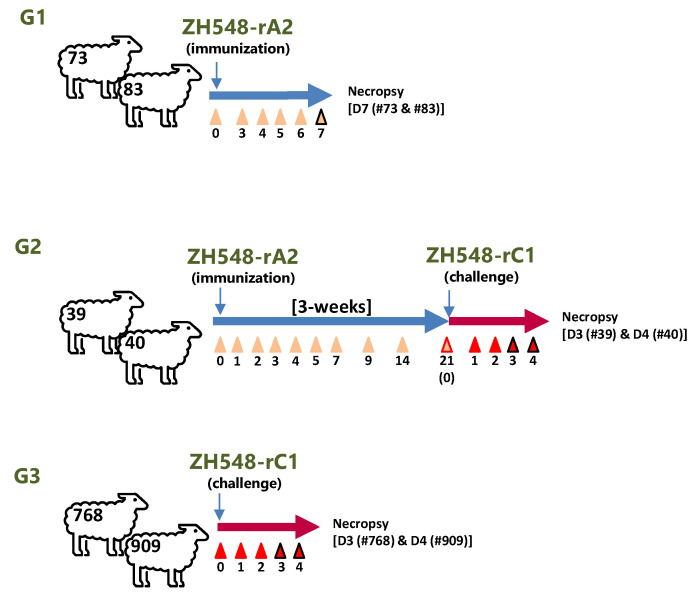
Experimental outline of the small-scale sheep trial described in this work. Three groups were defined for different aims. A first group (G1) was established for testing immunogenicity and pathology after rA2 virus inoculation. A second group (G2) was established to test efficacy of rA2 as a vaccine after rC1 challenge. The pathology of rC1 was tested in G3. Individual sheep numbers are indicated. Arrowheads indicate blood sample retrieval before (beige) or after challenge (red). Numbers below indicate days after immunisation or challenge. Blood samples were collected in EDTA tubes (for whole blood or plasma-based analysis) or in clotting tubes for serum retrieval.

**Figure 2 viruses-16-00087-f002:**
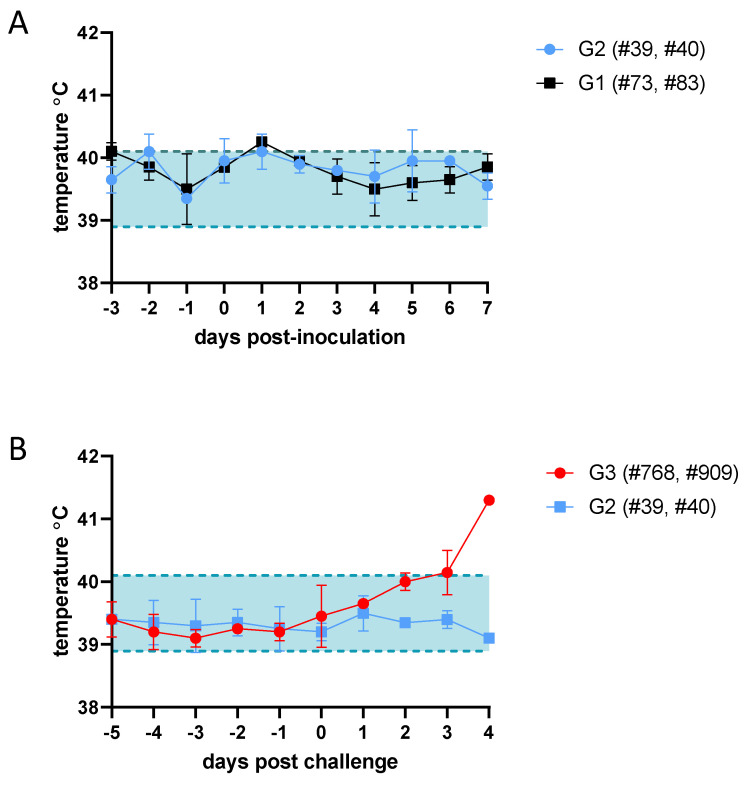
Daily mean rectal temperatures in rZH548-inoculated sheep. The plots represent mean and standard deviations. (**A**) Temperature records of the four sheep inoculated (vaccinated) with rA2 virus (G1 and G2) on similar days before and after inoculation (shown up to 7 dpi, as both sheep #73 and #83 were necropsied on this date). (**B**) Sheep temperatures upon rC1 challenge in both rA2-vaccinated (G2) and non-vaccinated (G3) sheep (for sheep #39 and #40, the −5 to −1 days correspond to 16 to 20 days post vaccination in graph A). The normal rectal temperature range for sheep is shaded between the dashed lines. Data in graphs available as Appendix A.

**Figure 3 viruses-16-00087-f003:**
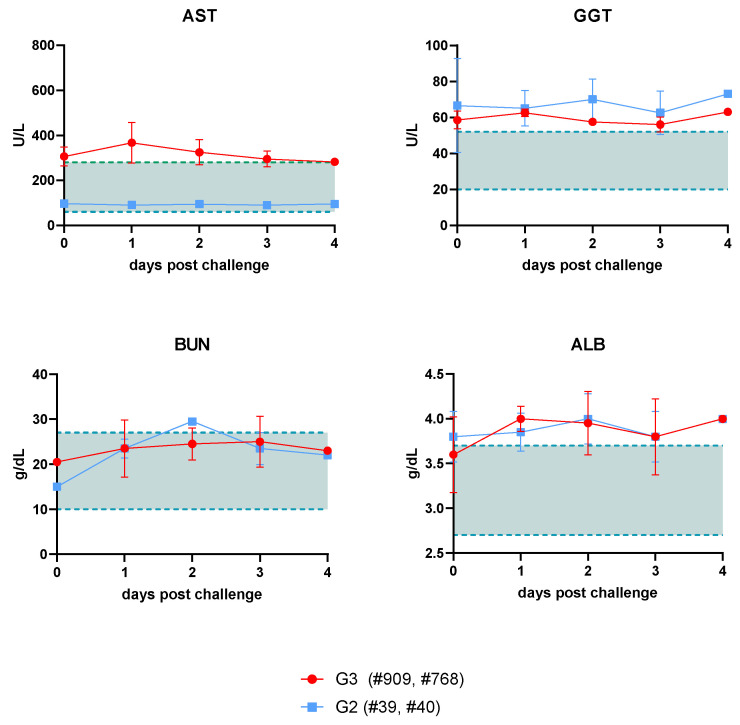
Blood chemistry analysis in vaccinated and/or challenged sheep (G2 and G3). Hepatic transaminases and protein degradation levels were measured in serum samples taken at the indicated days post challenge with rC1 virus. AST: aspartate aminotransferase. GGT: gamma-glutamyltransferase. ALB: serum albumin. BUN: serum urea nitrogen. The plots represent mean and standard deviations. Shaded areas denote normal enzyme values as described previously [32]. Data in graphs available as Appendix A.

**Figure 4 viruses-16-00087-f004:**
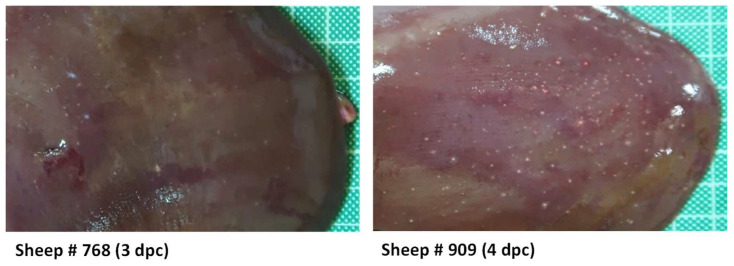
Macroscopic lesions in sheep liver. Representative images of livers taken during necropsies at days 3 (sheep 768) or 4 (sheep 909) post inoculation with rC1 virus (G3). Note the greyish-white foci, many of them surrounded by a reddish halo, scattered throughout the liver parenchyma, and corresponding to areas of necrosis.

**Figure 5 viruses-16-00087-f005:**
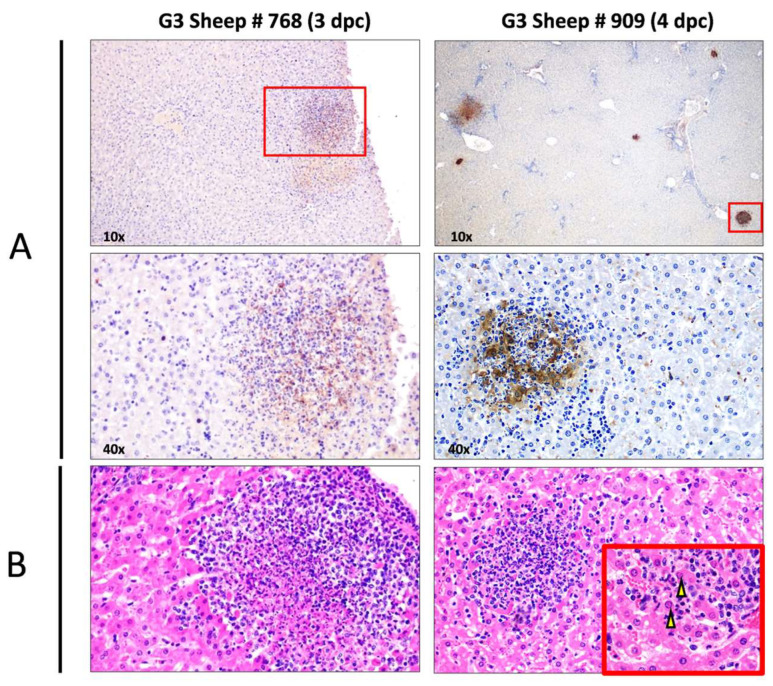
Histopathological and immunohistochemical findings observed in the liver of rC1-inoculated sheep (G3). (**A**) Detection of viral antigen in liver sections via immunohistochemistry using an in-house rabbit anti-RVFV polyclonal serum (upper panel magnification: 10×; lower panel magnification inset: 40×). Note the presence of immunolabelled hepatocytes within the necrotic foci, which were scattered throughout the liver parenchyma. (**B**) Haematoxylin–eosin stain. Necrotic foci filled with cellular debris and both degenerated and viable neutrophils, surrounded by mononuclear cells, mainly lymphocytes (magnification: 40×). Note the eosinophilic intranuclear inclusion bodies within the hepatocytes (inset arrows; magnification: 100×). Also, note the parallelism between viral antigen-immunolabelled foci and necrotic foci in haematoxylin–eosin sections.

**Figure 6 viruses-16-00087-f006:**
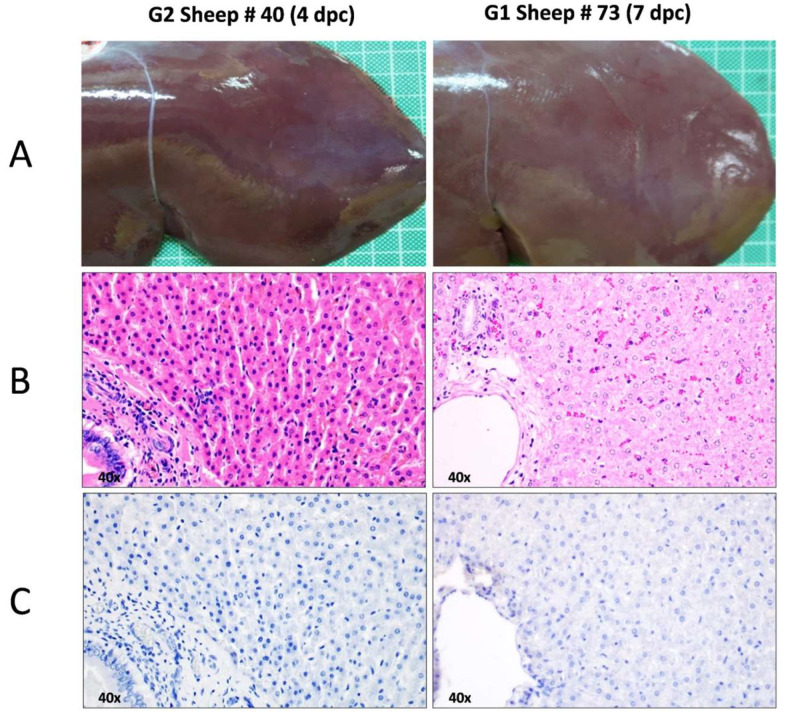
Macroscopic and histopathological evaluation of the liver of sheep inoculated with rA2 virus used as a vaccine candidate. (**A**) Representative macroscopic images of the liver of sheep #40 (inoculated with rA2 virus, challenged with rC1 virus and euthanised on day 4 post challenge) and sheep #73 (only inoculated with rA2 virus only and euthanised on day 7 post inoculation) to show the absence of macroscopic lesion. Serial sections of the liver of sheep #40 and #73 stained with haematoxylin–eosin (**B**) and immunolabelled against viral antigen with an in-house rabbit anti-RVFV polyclonal serum (**C**). Note the absence of liver lesions or cells marked against viral antigen (magnification: 40×).

**Figure 7 viruses-16-00087-f007:**
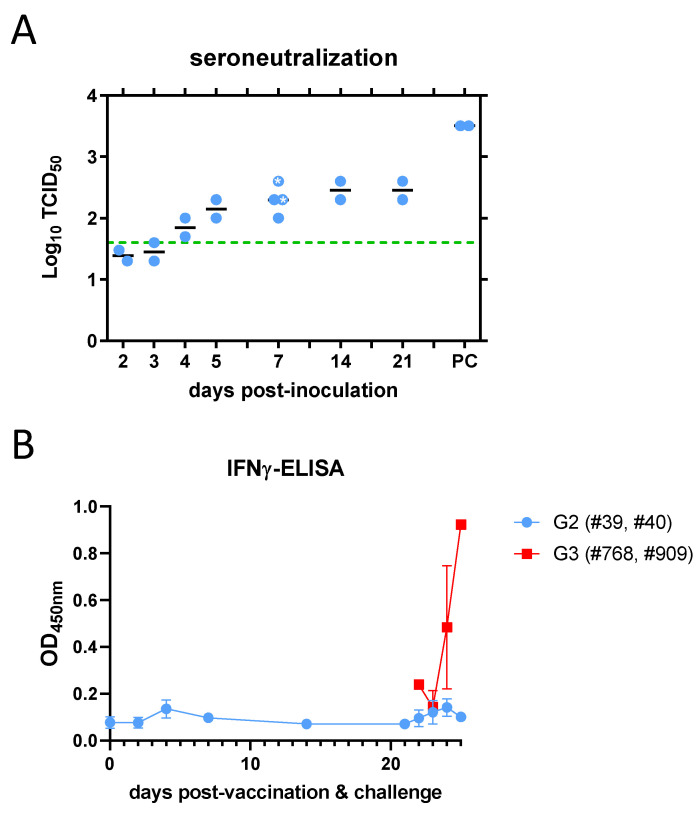
Analysis of humoral responses in sheep. (**A**) Kinetics of RVFV-neutralising antibody induction detected in serum microneutralisation test in rA2-inoculated sheep (G2). The measurement of neutralising antibodies in sheep #73 and #83 (G1) was performed only at day 7 post inoculation (symbols labelled with asterisk). Dotted line shows the predicted titre that correlates with protection of the assay. PC: post challenge (3 dpi). (**B**) Detection of interferon gamma in plasma samples by specific IFNγ capture sandwich ELISA. Challenge was performed at day 21. The plots represent mean and standard error. Data in graphs available as Appendix A.

**Figure 8 viruses-16-00087-f008:**
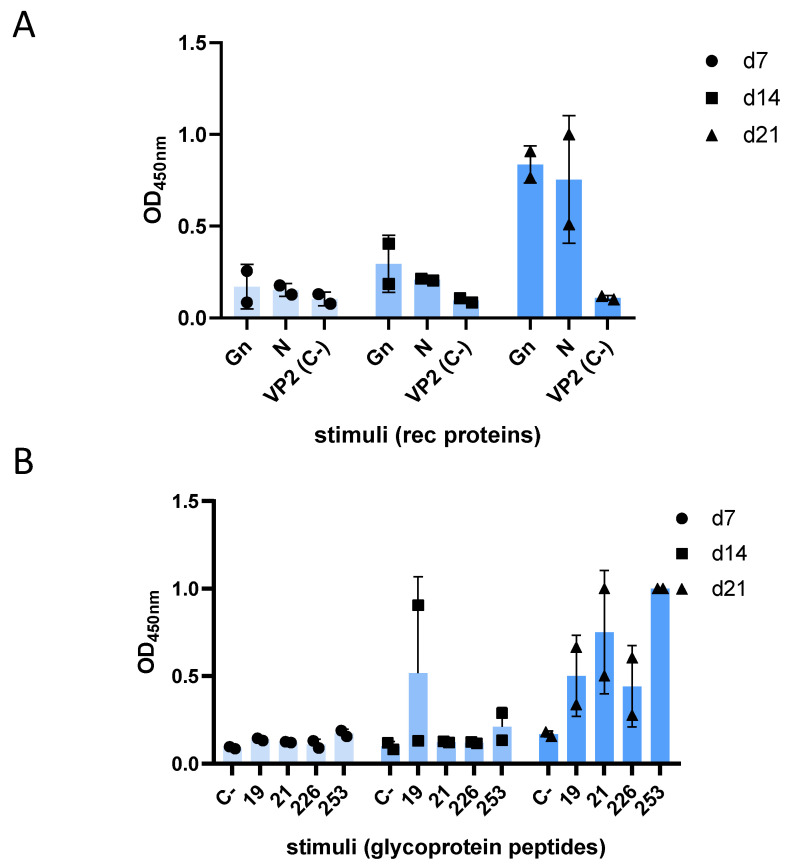
Determination of RVFV-specific T-cell responses in sheep. Detection of IFNγ in plasma from rA2-inoculated sheep (G2) by capture ELISA (**A**) Blood samples taken at 7, 14 or 21 days after rA2 inoculation were re-stimulated with recombinant RVFV Gn or N proteins. A recombinant VP2 protein from BTV-8 was used as a negative stimulus (C-). (**B**) Peptides derived from the Gn (#19 and #21) or Gc (#226 and #253) protein sequences were used for re-stimulation. The plots represent individual sheep values (symbols) and mean and standard deviations (bars). Data in graphs available as Appendix A.

## Data Availability

The data presented in this study are available in a Appendix A.

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
