# Peer review of "Safety and Efficacy upon Infection in Sheep with Rift Valley Fever Virus ZH548-rA2, a Triple Mutant Rescued Virus"

_viruses, 2024, doi:10.3390/v16010087_

Round 1
Reviewer 1 Report
Comments and Suggestions for Authors
Moreno et al submitted a manuscript titled "Safety and efficacy upon infection in sheep with Rift Valley fever virus ZH548-rA2, a triple mutant rescued virus" for publication in MDPI Viruses.
The work is interesting and well written. The virulence of an attenuated RVFV strain was rescued by introducing 3 mutations in RdRP and NSs.
There are no major issues, except,
add significance values in all graphs eg. fig. 2 and elsewhere), and
mention the stat tests used whenever significance values are mentioned in figure descriptions (eg. figs 7, fig. 8, and elsewhere).
Author Response
Moreno et al submitted a manuscript titled "Safety and efficacy upon infection in sheep with Rift Valley fever virus ZH548-rA2, a triple mutant rescued virus" for publication in MDPI Viruses.
The work is interesting and well written. The virulence of an attenuated RVFV strain was rescued by introducing 3 mutations in RdRP and NSs.
There are no major issues, except,
add significance values in all graphs eg. fig. 2 and elsewhere), and
mention the stat tests used whenever significance values are mentioned in figure descriptions (eg. figs 7, fig. 8, and elsewhere).
We perfectly understand this comment. We purposedly did not include statistics other than descriptive mean and deviations from the mean. The reason for this was obvious for us since in this experiment we set very small size groups (n=2) that makes not adequate to apply any parametric or non-parametric significance test. We agree this is a clear limitation of our work as we have stated in the discussion. Still, we believe that the individual outcomes observed can give a reasonable confirmation of what we observed in previous experiments in mice regarding the attenuation provoked by the introduction of three single mutations into the ZH548 genome. Our aim is not to pretend using ZH548-A2 as a vaccine but to show these positions as potential targets to add safety to attenuated vaccines. Therefore, performing an experiment with more sheep per group was not fully warranted, in our opinion, at least at this preliminary stage.
Nonetheless to be clearer for readers we have indicated in the graph’s captions the descriptive statistics (mean plus standard deviation or error) when necessary.
Reviewer 2 Report
Comments and Suggestions for Authors
Research by Moreno et al. presented here that the ZH548 strain bearing a triple mutation from the favipiravir resistant virus is highly attenuated on lamb, and confirmed the potential usage of this recombinant attenuated ZH548 strain as a live vaccine candidate. The overall quality is satisfactory, and the language is sufficient, however several minor amendments should be revised before publication.
Major comments:
The 5FU induced attenuated strain MP-12 is showing that still teratogenic in pregnant ewe. Whether this favipiravir attenuated RVFV strain still bearing this kind of feature should be discussed in the discussion part.
Minor comments:
Line 207-208, the antibody used in the experiment was antibody against bovine IFN-γ, as this manuscript was conducted on sheep, is this kind of antibody cross-species or a typo? Additionally, the cited ref is a review without detailed information about this kind of method and antibody.
Line 233, the corresponding figure results should be indicated within the sentence.
Line 260, the G2 related line seems not contain error bars, especially in AST, BUN related figures. This needs explanation. In addition, is the difference between the G2 and G3 groups significant? better indicate the P value, the differences in AST figure seem statistically positive.
Line 286 and Line 301, please indicate the corresponding group for each number of the lamb to ensure better understanding.
Line 472 the cited ref 20. was not updated.
Author Response
Research by Moreno et al. presented here that the ZH548 strain bearing a triple mutation from the favipiravir resistant virus is highly attenuated on lamb, and confirmed the potential usage of this recombinant attenuated ZH548 strain as a live vaccine candidate. The overall quality is satisfactory, and the language is sufficient, however several minor amendments should be revised before publication.
Major comments:
The 5FU induced attenuated strain MP-12 is showing that still teratogenic in pregnant ewe. Whether this favipiravir attenuated RVFV strain still bearing this kind of feature should be discussed in the discussion part.
We included this fact in the discussion, about the residual virulence of this strain, in order to show that its safety profile might be enhanced by introducing novel attenuating mutations such as those described in our work. The favipiravir attenuated virus (40Fp8) is only mentioned in this work as the original source for the three mutations used to attenuate rZH548 (40Fp8 carries additional mutations for which we still do not have data on attenuation).
Regarding teratogenicity of 40Fp8, we have just finished a trial in pregnant sheep for testing safety and efficacy of 40Fp8 as a vaccine which should be submitted soon, so we prefer not to anticipate these data in this article.
Minor comments:
Line 207-208, the antibody used in the experiment was antibody against bovine IFN-γ, as this manuscript was conducted on sheep, is this kind of antibody cross-species or a typo? Additionally, the cited ref is a review without detailed information about this kind of method and antibody.
This is correct. There is a cross-species reactivity, allowing the use of anti-bovine IFNg for sheep samples.
We have included the correct reference in which the protocol was originally detailed: Lorenzo et al. Priming with DNA plasmids encoding the nucleocapsid protein and glycoprotein precursors from Rift Valley fever virus accelerates the immune responses induced by an attenuated vaccine in sheep. Vaccine, Volume 26, Issue 41,2008, Pages 5255-5262.
Line 233, the corresponding figure results should be indicated within the sentence.
This has been corrected as suggested.
Line 260, the G2 related line seems not contain error bars, especially in AST, BUN related figures. This needs explanation.
The explanation for this is that the values were very similar in both G2 sheep, therefore the deviations are not perceptible in the graph. For AST this occurs at all days tested and for BUN in days 0 and 2 post challenge. In addition, there is only one value available at day 4 since one sheep on each group was sacrificed at day three. The numerical values have been supplied in a supplementary Excel file.
In addition, is the difference between the G2 and G3 groups significant? better indicate the P value, the differences in AST figure seem statistically positive.
As stated for the reviewer’s 1 response we have not applied statistical tests due to the very low size sample (of n=2) that we consider not enough to allow such analysis. However, we do not think these differences in AST values may be reflecting any response to the infection since the variations with respect to the levels detected at day 0 (before challenge) were minimal in both groups.
Line 286 and Line 301, please indicate the corresponding group for each number of the lamb to ensure better understanding.
This has been corrected as suggested in the corresponding figures.
Line 472 the cited ref 20. was not updated.
This reference has been now updated.
Reviewer 3 Report
Comments and Suggestions for Authors
Here is a two paragraph summary of the key points from the PDF:
This paper investigates the safety and efficacy of a genetically engineered, attenuated strain of Rift Valley fever virus (RVFV), called ZH548-rA2, as a vaccine candidate in sheep. The ZH548-rA2 virus contains three mutations in the viral polymerase and NSs virulence factor genes that were previously shown to fully attenuate the virus in mice. In this sheep trial, ZH548-rA2 caused no detectable clinical signs or pathology even at high doses, indicating it is highly attenuated compared to the parental virulent strain. Sheep inoculated with ZH548-rA2 produced neutralizing antibodies and cell-mediated immune responses. When these sheep were challenged with the virulent parental virus 3 weeks later, they were protected from developing liver lesions and showed no increase in liver enzymes, unlike control sheep inoculated only with the virulent virus strain which developed characteristic necrotic liver lesions.
In summary, this attenuated ZH548-rA2 RVFV strain appears to be both safe and immunogenic in sheep, inducing protective immunity against virulent RVFV challenge. The identification of key genomic sites that can attenuate RVFV without compromising immunogenicity provides valuable knowledge for improving the safety of live attenuated RVFV vaccines for livestock and potentially humans. Further studies with more animals are still needed to fully confirm the utility of this attenuated vaccine strain.
Minor Grammatical issue:
Line 183 Resombinant should be recombinant and three has an extra e.
Line 187 also has Concentrated misspelled.
Need to recheck the paper for spelling errors before publication.
The use of the word "Immunization" in reference to the ZH548-rA2 inoculation of the sheep may not be warranted given the small batch size and limitations of the experiments presented to date in this paper. It may be concluded that there was a difference seen in pathology seen in this short term experiment. I leave it to the authors if they want to draw that conclusion at this time.
Additional experiments:
Given the inability to detect virus in the blood, which is not entirely surprising given the often-transient nature of viral infections in the blood, it would be prudent if the samples exist, to test the liver tissue for attenuated virus and non-attenuated virus in the infected sheep.
Comments on the Quality of English LanguageMinor spelling mistakes that need to be corrected. Other than that, the Grammar is fine and the manuscript is well written.
Author Response
This paper investigates the safety and efficacy of a genetically engineered, attenuated strain of Rift Valley fever virus (RVFV), called ZH548-rA2, as a vaccine candidate in sheep. The ZH548-rA2 virus contains three mutations in the viral polymerase and NSs virulence factor genes that were previously shown to fully attenuate the virus in mice. In this sheep trial, ZH548-rA2 caused no detectable clinical signs or pathology even at high doses, indicating it is highly attenuated compared to the parental virulent strain. Sheep inoculated with ZH548-rA2 produced neutralizing antibodies and cell-mediated immune responses. When these sheep were challenged with the virulent parental virus 3 weeks later, they were protected from developing liver lesions and showed no increase in liver enzymes, unlike control sheep inoculated only with the virulent virus strain which developed characteristic necrotic liver lesions.
In summary, this attenuated ZH548-rA2 RVFV strain appears to be both safe and immunogenic in sheep, inducing protective immunity against virulent RVFV challenge. The identification of key genomic sites that can attenuate RVFV without compromising immunogenicity provides valuable knowledge for improving the safety of live attenuated RVFV vaccines for livestock and potentially humans. Further studies with more animals are still needed to fully confirm the utility of this attenuated vaccine strain.
Minor Grammatical issue:
Line 183 Resombinant should be recombinant and three has an extra e.
This has been corrected as suggested.
Line 187 also has Concentrated misspelled.
This has been corrected as suggested.
Need to recheck the paper for spelling errors before publication.
We have gone through the whole manuscript to revise spelling/grammar errors.
The use of the word "Immunization" in reference to the ZH548-rA2 inoculation of the sheep may not be warranted given the small batch size and limitations of the experiments presented to date in this paper. It may be concluded that there was a difference seen in pathology seen in this short term experiment. I leave it to the authors if they want to draw that conclusion at this time.
The term “immunization” is used here only when a further challenge was performed, considering that a previous inoculation would have “immunised” the sheep against a further virus exposure. We agree it may not be fully correct due to the limited sample size used, but we think it may be valid for easier understanding.
Additional experiments:
Given the inability to detect virus in the blood, which is not entirely surprising given the often-transient nature of viral infections in the blood, it would be prudent if the samples exist, to test the liver tissue for attenuated virus and non-attenuated virus in the infected sheep.
In our previous challenge experiment in sheep we usually were able to detect viral genomes in blood. We assumed this would be the case for rZH548. Thus, the lack of detection was unexpected. Indirectly, this was tested by immunohistochemistry (IHC), which indicates certain replication levels in positive samples. Unfortunately, we have no available frozen liver samples to test by RT-qPCR. One possibility would be to use deparaffined the liver samples used for histopathology and look there for viral genomes in order to have further confirmation but looking at the IHC data we are confident to suggest that in sheep, when compared to the A2 mutant virus, the rZH548 virus is able to replicate to some extent.